# Changes in Internal Structure and Dynamics upon Binding Stabilise the Nematode Anticoagulant NAPc2

**DOI:** 10.3390/biom14040421

**Published:** 2024-03-30

**Authors:** Elaine Woodward, Brendan M. Duggan

**Affiliations:** Department of Biochemistry and Molecular Biology, Medical University of South Carolina, Charleston, SC 29425, USA

**Keywords:** nematode, anticoagulant, molecular dynamics, factor Xa, salt bridge, protease inhibitor

## Abstract

Abnormal blood coagulation is a major health problem and natural anticoagulants from blood-feeding organisms have been investigated as novel therapeutics. NAPc2, a potent nematode-derived inhibitor of coagulation, has an unusual mode of action that requires coagulation factor Xa but does not inhibit it. Molecular dynamics simulations of NAPc2 and factor Xa were generated to better understand NAPc2. The simulations suggest that parts of NAPc2 become more rigid upon binding factor Xa and reveal that two highly conserved residues form an internal salt bridge that stabilises the bound conformation. Clotting time assays with mutants confirmed the utility of the salt bridge and suggested that it is a conserved mechanism for stabilising the bound conformation of secondary structure-poor protease inhibitors.

## 1. Introduction

Thrombosis, the abnormal formation of a blood clot within the circulatory system and the subsequent blockage of blood flow, affects millions of people every year. Deep vein thromboses and pulmonary embolisms are responsible for at least 100,000 deaths annually in the US and the incidence is suspected to be significantly under-reported [1]. Prevention of thrombosis has been described as the single most important way to reduce hospital deaths. The two major pharmaceutical agents for the prevention of thrombosis are heparin and warfarin, but both have significant drawbacks and alternative anticoagulants have been sought [2].

Hematophagous organisms need to keep blood flowing to feed and offer a rich source of anticoagulants. The nematode *Ancylostoma caninum* obtains its blood meal from lacerations it creates, primarily in the small intestine of dogs, and produces several different anticoagulant proteins [3]. These nematode anticoagulant proteins (NAPs) are small (approximately 9 kDa), acidic proteins that contain ten conserved cysteine residues paired to form five disulfide bridges. These NAPs are some of the most potent natural anticoagulants yet discovered. NAP5 has a 43 pM inhibitory constant for activated coagulation factor X (fXa), while NAPc2 exhibits an 8.4 pM inhibitory constant for activated coagulation factor VII (fVIIa) in complex with Tissue Factor (TF) [3], and has entered multiple clinical trials [4]. NAP5, NAP6, and NAP12 inhibit fXa by binding to the protease’s active site [3,5], while NAPc2, NAPc3, and NAPc4 bind to an exosite on fXa and, in a quaternary complex, inhibit fVIIa when bound to TF [2,6] (Figure 1).

NMR solution studies of NAPc2 found the anticoagulant has little regular secondary structure and extensive, disordered, highly mobile regions [7]. A crystal structure of NAPc2 in complex with fXa revealed that the mobile carboxyl terminus of NAPc2, disordered when free, forms a structured interaction face between the two proteins, extending an existing β-sheet in fXa [6]. Unfortunately, much of the rest of NAPc2 was too disordered to be seen in the crystal structure. To provide insight into the unusual inhibitory mechanism of NAPc2, we carried out all-atom simulations of NAPc2, fXa, and the NAPc2-fXa complex in explicit solvent. The simulations suggested the anticoagulant was more stable when bound to the protease and that an internal salt bridge between highly conserved residues may stabilize the bound conformation. Mutants of NAPc2 and NAP5 that could not form the salt bridge were found to be less potent inhibitors of coagulation than the wild-type proteins, supporting the importance of the salt bridge for the function of the inhibitors.

## 2. Materials and Methods

### 2.1. Molecular Dynamics Simulations

MD simulations were generated using GPU-enhanced versions [8,9,10] of the Amber suite of programs [11] and the ff14SB forcefield [12]. All simulations used the particle mesh Ewald procedure [13] under periodic boundary conditions. A time step of 2 fs was used for all simulations and the SHAKE algorithm [14] was used to constrain the length of bonds to hydrogen atoms. All-atom representations of the molecules were created from PDB files 1COU [7] and 2H9E [6]. The fXa structures consisted of two peptide chains. The heavy chain residues were numbered consecutively from 1 to 233 and included the serine protease domain. The light chain residues were numbered from 234 to 281 and included an EGF domain. The heavy and light chains are linked by a disulfide bridge. Missing hydrogen atoms were placed and protonation states were determined using reduce [15]. Rotamers of missing sidechains were selected from the most populated rotamers in the 2010 Dunbrack library [16]. NAPc2 atoms missing from the NAPc2-fXa structure were modelled based on the lowest energy NAPc2 NMR structure (the first). The proteins were solvated in truncated octahedral boxes of TIP3P water [17]. The charge was neutralised by the addition of sodium and chloride ions to give a concentration of 150 mM NaCl [18]. The characteristics of the simulation systems are summarised in Table 1.

The solvated systems were prepared for production through multiple energy minimisation and relaxation steps as follows. First, the solvent was minimised with 20 steps of steepest descent energy minimisation followed by 980 steps of conjugate gradient minimisation with positional restraints holding all protein atoms fixed. The second stage minimised the entire system with 100 steps of steepest descent followed by 4900 steps of conjugate gradient minimisation. Next, the systems were heated from 100 K to 310 K over 1 ns of NVT molecular dynamics with all protein atoms restrained. The systems were then allowed to relax at a constant pressure of 1 atm under Langevin dynamics [19] with a collision frequency of 1 ps^−1^ while the positional restraints on the protein atoms were gradually reduced and finally removed.

Production runs of 100 ns were performed in the NPT ensemble at 1 atm using the Berendsen barostat [20] and 310 K using the Langevin thermostat. Co-ordinates were saved every 0.1 ns. Trajectories were analysed using cpptraj [21]. The secondary structure was defined using the DSSP method [22]. Phi torsion angles were converted to ^3^*J*_HNHα_ coupling constants using the Karplus relationship with parameters A = 7.13, B = −1.31 and C = 1.56 [23,24]. Dynamic cross-correlation matrices [25] were calculated using Cα atoms. Salt bridges required a heavy atom distance of <4.0 Å [26]. Data was plotted with GnuPlot (www.gnuplot.info, version 4.6.2), structures were visualized with UCSF Chimera (www.cgl.ucsf.edu/chimera, version 1.17.3) [27], and figures were assembled with GIMP (www.gimp.org, version 2.8.22).

### 2.2. Plasmid Construction

The NAP5 sequence was PCR amplified from an adult *A. caninum* cDNA library that was a generous gift from Professor Peter Hotez of George Washington University. The primers used were 5′-GAA AAC CTG TAC TTT CAG GGA AAG GCA TAC CCG GAG TGT GGT GAG-3′ and 5′-TCA GAC ATG TAT AAT CTC ATG TTG GTC GCA TTC TTC TTC C-3′ and were purchased from Integrated DNA Technologies (Coralville, IA, USA. Attempts to isolate NAPc2 cDNA from the adult *A. caninum* cDNA library were unsuccessful. Instead, the sequence was constructed by PCR using long overlapping primers. The primers used were 5′-GAA AAC CTG TAC TTT CAG GGA AAA GCA ACG ATG CAG TGT GGT GAG AAT GAA AAG TAC GAT TCG TGC GGT AGC AAG GAG TGC GAT AAG AAG TGC AAA TAT GAC GGA GTT GAG GAG GAA GAC GAC GAG GAA CCT AAT GTG CCA TGC CTA GTA CGT GTG TGT CAT CAA GAT TGC GTA TGC G-3′ and 5′-CGG AGT TGA GGA GGA AGA CGA CGA GGA ACC TAA TGT GCC ATG CCT AGT ACG TGT GTG TCA TCA AGA TTG CGT ATG CGA AGA AGG ATT CTA TAG AAA CAA AGA TGA CAA ATG TGT ATC AGC AGA AGA CTG CGA ACT TGA CAA TAT GGA CTT TAT ATA TCC CGG AAC TCG AAA CCC TTG A-3′. The PCR products were used to create Gateway entry vectors by TOPO-TA cloning with the pCR8/GW/TOPO vector (Invitrogen, Carlsbad, CA, USA). Recombination with a Gateway p(His6)-GB1-GTW destination vector produced a construct encoding an N-terminal His6 tag for purification, a GB1 domain for enhanced expression, a tobacco etch virus protease cleavage site, and the desired nematode anticoagulant protein. E10Q mutants were produced by QuickChange (Agilent, Santa Clara, CA, USA) reactions.

### 2.3. Protein Expression and Purification

*Escherichia coli* BL21-CodonPlus (DE3)-RIPL competent cells (Agilent, Santa Clara, CA, USA) were transformed with plasmids expressing the His_6_-GB1-NAP constructs and expression was induced with 0.5 mM isopropyl β-D-thio-galactoside for 2–4 h at 37 °C. The protein was isolated by chromatography on a HisTrapFF affinity column (GE Healthcare, Piscataway, NJ, USA). On-column refolding with urea in the presence of dithiothreitol was used to ensure correct disulfide pairing. Cleavage with tobacco etch virus protease removed the His_6_ tag and GB1 domain and released the NAP from the column. NAPs were exchanged into a 150 mM sodium chloride, 10 mM sodium phosphate pH 7.4 buffer, using a PD-10 desalting column (GE Healthcare, Piscataway, NJ, USA). Mass spectrometry and NMR confirmed that the correct proteins were produced and that wild-type and mutant anticoagulants were correctly folded.

### 2.4. Prothrombin Time Coagulation Assays

Prothrombin time coagulation assays were run in triplicate at 37 °C in 96-well plates. Normal human plasma (George King Biomedical, Overland Park, KS, USA) was thawed immediately before use and incubated with buffer or anticoagulant for 15 min before initiating coagulation with thromboplastin (George King Biomedical, Overland Park, KS, USA). A kinetic plate reader was used to record UV absorbance at 650 nm every six seconds for 5 min. Plotting the absorbance data against time gave a sigmoidal curve whose inflection point was taken as the coagulation time.

## 3. Results

### 3.1. Simulations Capture Mobility of NAPc2

The root mean square deviations (RMSDs) of each of the production runs are shown in Figure 2A,B. While the fXa RMSDs are low and remain stable throughout the entire trajectory, the RMSDs of NAPc2 are high in both the free and bound simulations, suggesting the NAPc2 simulations had not stabilised. The energies of all three simulations (Figure 2C), however, remain almost constant throughout the simulations.

To determine if the simulations provided suitable data for further analysis, the secondary structure throughout the trajectories was examined (Appendix A). In general, the secondary structure of both proteins remains the same throughout the production run. Some residues fluctuate between α-helix, 3_10_-helix, and turn, and some residues form transient β-strands; however, these are fluctuations about a consensus structure rather than a drift towards another state. The β-strand formed by NAPc2 residues M75-I78 to bind to fXa is present throughout the entire production simulation. The consistency of the secondary structure suggested that the entire production run could be used for analysis, as it provides a description of the conformational sampling undertaken by the proteins.

As a further test of the validity of the NAPc2 simulation, experimental NMR data [7] was compared with data derived from the free NAPc2 simulation. The torsion angle φ at each frame of the simulation was used to derive ^3^*J*_HN,Hα_ coupling constants then averaged for comparison with experimental values (Appendix A). Additionally, atomic positional fluctuations (APFs) from the simulation were compared with {^1^H}^15^N heteronuclear NOEs (Appendix A). The coupling constants derived from the simulation are in the same range as those measured experimentally (Appendix A), but the correlation is poor (Appendix A), other than for residues C70-Y79 that encompass the only α-helix in NAPc2. The APFs show similar trends to the heteronuclear NOE. The termini and the acidic region in the centre of the NAPc2 sequence (E30-E36) both show increased mobility, which results in high APFs and reduced heteronuclear NOEs (Appendix A). Again, the correlation is poor (Appendix A), but the broad trends are consistent. Despite the high RMSDs shown by NAPc2 in the free and fXa-bound simulations, the conservation of secondary structure, the general agreement with experimental data, and the stability of the total energy suggest that the simulations reproduced the inherent mobility of NAPc2 and can be used for further analysis to understand its inhibitory mechanism.

NMR data to validate the fXa simulation was not available, but previous fXa simulations had reported aromatic residues acting as a gate to the active site [28]. Visual inspection of selected structures showed these aromatic side chains flipping back and forth (Appendix A). Measuring the distances between these two residues in the simulation revealed large-scale movement over the course of the simulation (Appendix A). In addition, the distance between a bound sodium ion and its neighbouring residue showed transitions between two preferred states (Appendix A). Taken together these observations indicate that the fXa structure was stable over the course of the simulation and that the simulation reproduced previously reported dynamic behaviour of the protein.

### 3.2. The fXa Bound Conformation Was Not Present in the Free NAPC2 Simulation

The crystal structure of the NAPc2-fXa complex revealed that the flexible carboxyl terminus of NAPc2 forms a new β-strand that packs against an existing β-sheet in fXa [6]. The ensemble of NAPc2 NMR structures shows no evidence of the β-strand (Appendix A), so it was of interest to see if this conformation is present in the free NAPc2 simulation and thus determine whether the interaction is an example of conformational selection, rather than induced fit. In the 100 ns production run, the RMSD between the β-strand forming residues (M75-I78) in the free NAPc2 simulation and those residues in the fXa-bound structure remained more or less constant at 3.2 Å (Figure 3B), suggesting the bound conformation is not present in this free NAPc2 simulation. Superimposing the β-strand residues from the free NAPc2 simulation with the least, greatest, and mean RMSDs on the same residues in the bound structure (Figure 3A) shows that in the free NAPc2 simulation, the backbone atoms of D76 and F77 adopt a different conformation. The absence of the bound conformation from the free simulation does not prove that the bound conformation is induced upon binding; it merely shows that this simulation did not sample the bound conformation. It is possible that more extensive simulations may sample the bound conformation, but with the data available here the question of conformational selection versus induced fit cannot be answered.

### 3.3. Binding to fXa Stabilizes NAPc2

The secondary structure analysis showed that the NAPc2 β-strands and α-helix were all extended slightly when bound to fXa (Appendix A), suggesting a more stable structure. Dynamic cross-correlation analysis shows that the bound form of NAPc2 has more extensive regions of correlated motions than the free (Figure 4). In the free NAPc2 simulation (Figure 4A), residues C52-D73 show correlated motions, as do residues D76-P85. These two groups show some anti-correlated motions, indicating that residues N74 and M75 act as a hinge between the two groups. Residues near the cysteines also show correlated motions. In the NAPc2-fXa simulation (Figure 4B), residues K1-D27 and E32-M75 are all extensively correlated. Residues 76–85 show no correlation with the rest of the protein, indicating that, when bound to fXa through the C-terminal residues, the rest of NAPc2 moves as a stable unit, independent of the C-terminus. Applying the same analysis to fXa revealed few correlated motions in either the free or bound forms and little difference between the two (Appendix A).

### 3.4. Internal Salt Bridge Found in Bound Conformation

Examination of the stabilised core residues in the NAPc2-fXa simulation revealed the presence of a salt bridge between residues E10 and R58 (Figure 5). This salt bridge was present for 70.2% of the NAPc2-fXa simulation but was not observed at all in the NAPc2 simulation. The salt bridge was present at the start of the NAPc2-fXa production run, but not in the initial structure of the complex, which was modelled on the NAPc2-fXa crystal structure [6] where the R58 side chain was not observed. Towards the end of the equilibration simulations, the salt bridge was formed). Residues E10 and R58 are highly conserved in the nematode anticoagulants and other members of the trypsin inhibitor-like cysteine-rich family (Appendix A), suggesting an evolutionarily conserved mechanism for stabilising the bound conformation [17].

### 3.5. Internal Salt Bridge Required for Maximal Activity

To test if the salt bridge observed in the NAPc2-fXa simulation was required for inhibitory activity, wild-type and E10Q mutants of NAPc2 and NAP5 were prepared and tested in coagulation assays. NMR analysis of the expressed proteins indicated that they were folded and that the E10Q mutation made no appreciable difference to the structure (Appendix A). The E10 sidechain is exposed to the solvent, and so the mutation is unlikely to disrupt the structure. The wild-type proteins inhibited coagulation, but the E10Q mutants were much less effective (Figure 6), confirming that the internal salt bridge was required for maximal activity.

## 4. Discussion

The hookworm *Ancylostoma caninum* produces numerous anticoagulant proteins. The most extensively investigated, NAPc2, is the most potent natural inhibitor of the fVIIa-TF complex yet discovered and has entered clinical trials for the treatment of venous and arterial thrombosis [4]. NAPc2 inhibits the fVIIa-TF complex, but requires the presence of fXa and phospholipid membranes [3]. A mechanism reconciling these facts proposes that NAPc2 binds to fXa at an exosite and uses the affinity of fXa for the fVIIa–TF complex to deliver it to fVIIa, its true target (Figure 1). Previously, NMR studies of NAPc2 revealed that it has little regular secondary structure and extensive, highly mobile, disordered regions [7]. A crystal structure of NAPc2 bound to fXa confirmed that the anticoagulant did not occupy the active site of fXa and remained disordered when bound [6]. In an effort to better understand the unusual mechanism of NAPc2, we generated molecular dynamics simulations of NAPc2, fXa, and the NAPc2-fXa complex.

The NAPc2 simulations showed unusually high mobility of the anticoagulant in both free and fXa-bound states (Figure 2A,B). However, this was consistent with NMR observables (Appendix A) and the inability to observe all of the anticoagulant in the crystal structure of the complex. The fXa simulations were much more stable and reproduced previously reported behaviour (Appendix A). This led us to conclude useful information could be derived from the simulations.

The previously reported NMR structure of NAPc2 was highly disordered but showed no evidence of the β-strand that forms the interface in the NAPc2-fXa crystal structure (Appendix A), raising questions about how the complex forms. Searching for the bound conformation in the free simulation failed to find a close match (Figure 3), which unfortunately fails to answer the question of whether or not the extensive mobility in the free protein is sufficient to allow it to sample the bound conformation before binding.

In the simulations, the limited secondary structure of NAPc2 was observed to extend when bound to fXa (Appendix A), and extensive correlated motions were detected (Figure 5). These observations suggest that upon binding to fXa, the original NAPc2 secondary structure regions become more extensive and stable, forming a more rigid core. Hinge residues between the core and the anchored carboxyl terminus allow the bound NAPc2 to sample a range of space. Importantly, this leaves the cleavage loop residues, which inhibit the active site of fVIIa, projecting away from fXa and allowing NAPc2 a range of motion that can accommodate dynamic reorganisation to inhibit the fVIIa-TF complex (Figure 7). Interestingly, the amino acid sequences of the majority of *A. caninum* NAPs are more like NAPc2 than NAP5 (Appendix A), suggesting that most of these anticoagulants use the same mechanism as NAPc2, i.e., targeting the fVIIa-TF complex rather than fXa.

The MD simulations revealed an internal salt bridge between two highly conserved residues that could help stabilise the core of NAPc2 when bound to fXa (Figure 5). To test if the salt bridge contributed to the anticoagulant activity, one of the residues was mutated to prevent the formation of the salt bridge. Clotting time assays showed that this mutation greatly reduced the potency of NAPc2 and of another nematode anticoagulant, NAP5 (Figure 6). The glutamic acid and arginine residues that form the salt bridge are highly conserved across the entire TIL (Trypsin Inhibitor-Like cysteine-rich) family (Appendix A). In some cases, the conserved residues form a salt bridge (CE-1 bound to porcine elastase [29]), while in others it does not (NAP5 bound to fXa [30], ATI [31], and AMCI-1 [32]). Formation of the salt bridge is likely to stabilise these secondary structure-poor protease inhibitors, and where these residues are missing [33], their introduction may improve their potency.

## 5. Conclusions

Molecular dynamics simulations of the nematode anticoagulant NAPc2 and the protease that it binds to, fXa, have provided insight into the dynamics and mode of action of this potent anticoagulant, suggesting how it may inhibit the fVIIa-TF complex. The simulations also suggested a purpose for a pair of the highly conserved residues in this family of protease inhibitors, a purpose that was confirmed by constructing mutants and demonstrating their decreased potency.

## Figures and Tables

**Figure 1 biomolecules-14-00421-f001:**
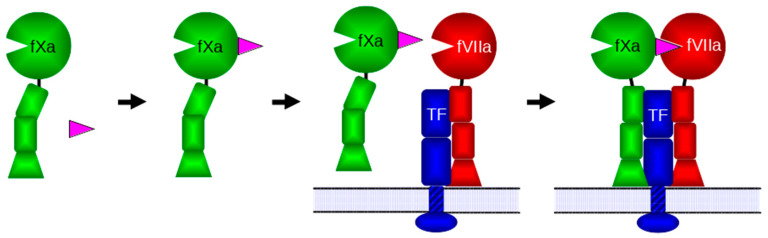
NAPc2 inhibition of fVIIa. NAPc2 is shown in magenta, fXa in green, fVIIa in red and TF in blue. In fXa and fVIIa circles represent serine protease domains, rectangles EGF domains and trapezoids γ-carboxyglutamate (Gla) domains. Tissue factor is shown embedded in a lipid membrane via a transmembrane domain (striped rectangle). The rectangular TF domains are fibronectin type III modules and the oval is the cytoplasmic signalling domain. NAPc2 binds to the fXa serine protease domain without blocking the active site. The NAPc2–fXa complex binds to the fVIIa-TF complex on the cell surface using interactions between the fXa Gla domain and the phosphatidylserine head groups of the membrane, thereby delivering NAPc2 to fVIIa where it can block the active site of the protease.

**Figure 2 biomolecules-14-00421-f002:**
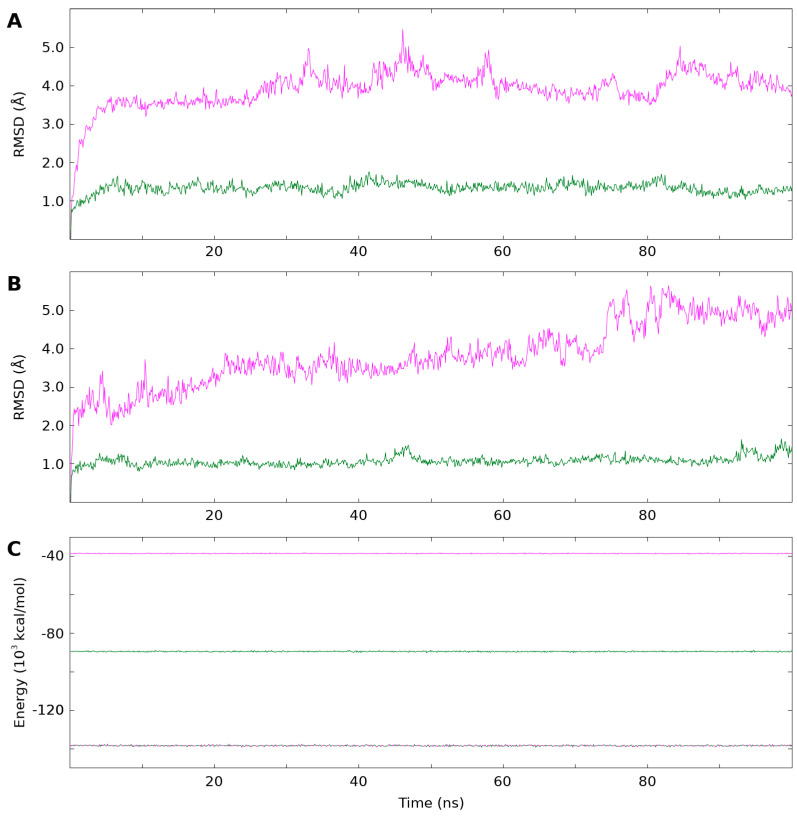
RMSDs and total energy during the three simulations. RMSDs of NAPc2 are shown in magenta and of fXa in green. RMSDs were calculated by fitting the backbone heavy atoms to the initial frame of each trajectory. (**A**) RMSDs of the free proteins. The mean RMSD of free NAPc2 was 3.9 ± 0.5 Å and of free fXa 1.4 ± 0.1 Å (**B**) RMSDs of the bound proteins. The mean RMSD of bound NAPc2 was 3.8 ± 0.8 Å, and of bound fXa 1.1 ± 0.1 Å (**C**) Total energy of NAPc2 in magenta, fXa in green, and the NAPc2-fXa complex in dashed magenta and green. The mean total energy of NAPc2 was − 38,500 ± 100 kcal/mol, of fXa − 89,500 ± 200 kcal/mol, and of the NAPc2-fXa complex − 138,400 ± 300 kcal/mol.

**Figure 3 biomolecules-14-00421-f003:**
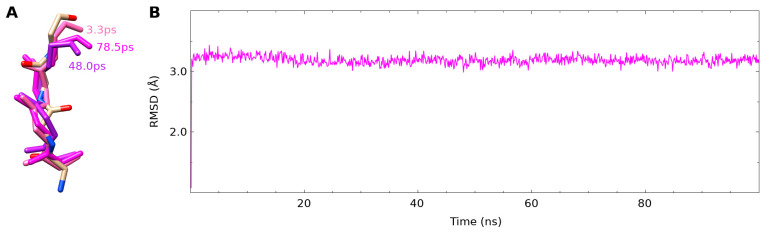
RMSDs of NAPc2 interface residues in free and bound simulations. (**A**) Selected frames from the NAPc2 simulation are superimposed on the backbone atoms of the residues forming the NAPc2 β-strand (M75-I78) that binds to fXa in the NAPc2-fXa complex. Backbone atoms in the crystal structure of the NAPc2-fXa complex (2H9E) are shown in CPK colour scheme. The closest frame from the NAPc2 simulation is shown in purple (78.5 ps, RMSD = 2.99 Å), the furthest frame in pink (3.3 ps, RMSD = 3.44 Å), and a frame close to the mean in magenta (78.5 ps, RMSD = 3.19 Å). (**B**) RMSD of M75-I78 backbone heavy atoms in the NAPc2 simulation to the same atoms in the first frame of the NAPc2-fXa simulation plotted against time. Excluding the first frame with an RMSD of 1.08 Å, the mean RMSD was 3.19 ± 0.07 Å.

**Figure 4 biomolecules-14-00421-f004:**
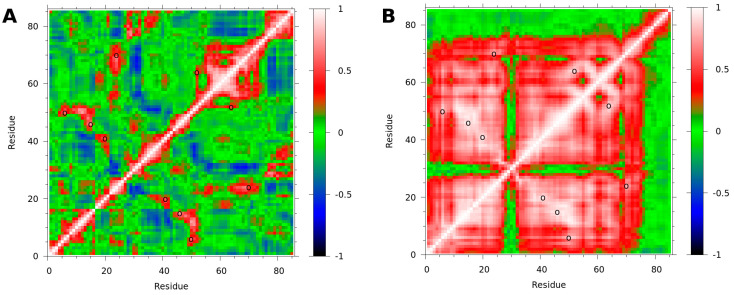
Correlation analysis of NAPc2 in free and bound simulations. Dynamic cross-correlation matrices were calculated using Cα atoms. A correlation of 1.0 (red) indicates correlated motions, i.e., the atoms move in the same direction. A correlation of −1.0 (blue) indicates anti-correlated motions, i.e., the atoms move in opposite directions. A correlation of 0.0 (green) indicates no correlated motions. Open circles indicate disulfides. (**A**) Correlation matrix of NAPc2 from the NAPc2 simulation. (**B**) Correlation matrix of NAPc2 from the NAPc2-fXa simulation.

**Figure 5 biomolecules-14-00421-f005:**
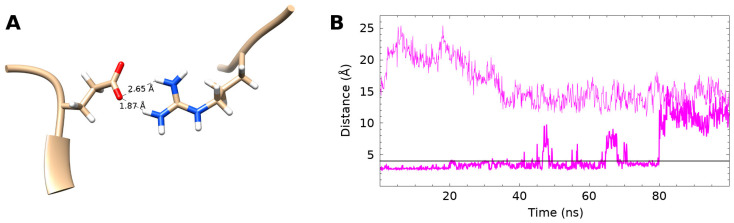
Internal salt bridge found in the NAPc2-fXa simulation. (**A**) Residues E10 and R58 at 20.0 ps in the NAPc2-fXa simulation are shown in CPK colour scheme. Numbers indicate the distance between the E10 oxygen and R58 hydrogen atoms. (**B**) The distance between the E10 Oε and R58 Nη atoms plotted against simulation time. The upper, light magenta line is from the NAPc2 simulation while the lower, heavy line is from the NAPc2-fXa simulation. The horizontal black line at 4.0 A indicates the cutoff below which a salt bridge is considered to be present. In the NAPc2 simulation, the salt bridge did not form, whereas in the NAPc2-fXa simulation, the salt bridge was present 70.2% of the time. When the salt bridge was formed, the mean distance between the E10 Oε and R58 Nη atoms was 3.2 ± 0.4 Å. In the free NAPc2 simulation, the mean distance between the E10 Oε and R58 Nη atoms was 16 ± 3 Å.

**Figure 6 biomolecules-14-00421-f006:**
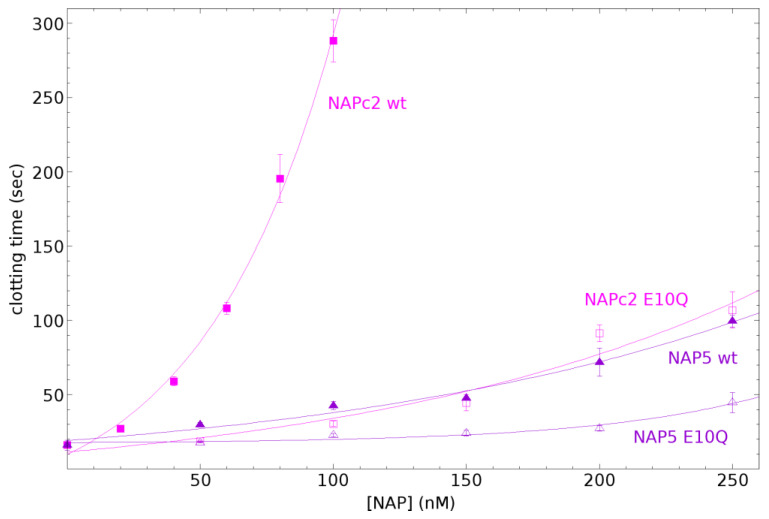
Inhibition of coagulation by wild-type NAPc2 and NAP5 and their E10Q mutants. Clotting times for wild-type NAPc2 are shown in filled magenta squares, for E10Q NAPc2 in open magenta squares, for wild-type NAP5 in filled purple triangles, and for E10Q NAP5 in open purple triangles. Error bars are the standard deviation of at least three measurements.

**Figure 7 biomolecules-14-00421-f007:**
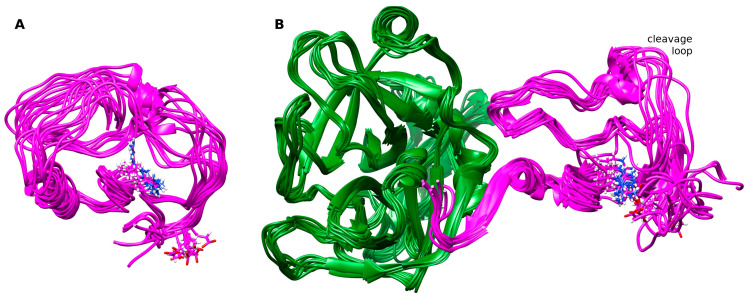
Representative structures from the simulations. Structures were taken every 10 ns from (**A**) the NAPc2 and (**B**) NAPc2-fXa simulations. NAPc2 is shown in magenta with the E10 and R58 sidechains in stick form. fXa is shown in green with the serine protease domain in front and the EGF domain at the rear. Structures from the NAPc2-fXa simulation were superimposed on the fXa backbone atoms. Structures from the NAPc2 simulation were superimposed on the NAPc2-fXa structures using the backbone atoms of rigid NAPc2 residues (E8-C24, C46-M75), then translated to the left. The location of the NAPc2 cleavage loop that occupies and inhibits the active site of fVIIa is indicated. Notice how the disordered NAPc2 C-terminus forms a β-strand to bind to fXa and much of the NAPc2 structure becomes more ordered.

**Table 1 biomolecules-14-00421-t001:** Characteristics of the systems used for molecular dynamics simulations.

	NAPc2	fXa	NAPc2-fXa
PDB code	1COU ^1^	2H9E	2H9E
PDB method	NMR	X-ray	X-ray
residues	85	281	366
protein atoms	1267	4312	5579
counterions	18 Na^+^, 6 Cl^−^	28 Na^+^, 29 Cl^−^	63 Na^+^, 50 Cl^−^
water molecules	4481	10,434	15,885
box size (Å)	33 × 28 × 44	47 × 65 × 53	53 × 57 × 80
simulation length (ns)	100	100	100

^1^ The lowest energy NMR structure (the first in the ensemble) was used.

## Data Availability

The data presented in this study are available on request from the corresponding author.

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
