# Peer review of "Changes in Internal Structure and Dynamics upon Binding Stabilise the Nematode Anticoagulant NAPc2"

_biomolecules, 2024, doi:10.3390/biom14040421_

Round 1

Reviewer 1 Report (Previous Reviewer 4)

Comments and Suggestions for Authors

Comments on biomolecules-2940898:

The current paper as a majorly revised version of the previous submission improves in various aspects, including changing the force-field selection from the old 99SB that exhibits certain limitations in e.g., helix propensities to a more modern-level selection 14SB, adding statistical uncertainties to various statistical observables extracted from simulations, thoroughly text editing removing many typos and duplicate definitions of abbreviations, increasing the sampling interval to decorrelate snapshots and many others, which align with my suggestions on the previous submission.

However, despite these improvements, a worry about the scientific content still arises. Throughout the paper, the comparison between experiment and simulation outcome is rather limited. For the existing J-coupling comparison, the consistency is rather poor. This could be caused by the intrinsic limitation of the method or the inaccurate modelling details used in simulations, but the exact reason remains unknown without further detailed investigations. We normally expect more meaningful scientific findings at the all-atom level to be reported in a modelling paper, e.g., binding energetics or conformational changes such as relative stabilities of different conformations. I’m not sure whether this is acceptable for the journal, and I would leave the question to the other reviewers and editors and suggest minor revision. Below are additional detailed comments on the new version.

The energetics reported in Figure 2c are not really useful. The total energy of the system would not vary monotonically with an acceptable initial condition. Further, the huge statistical uncertainty +-200 kcal/mol makes the estimates statistically not really meaningful.

With modern computational resources, doing a 1000 ns simulation for protein systems (even large biomolecular assemblies) is rather easy. As a result, the current ~100 ns sampling seems slightly short in modern simulation works.

The RMSD analysis in Figure 3 may not be a useful method to quantify the structural variation in this region. Its value varies severely in the first several frames and remains ~3 Angstrom afterwards. Perhaps changing to other structural descriptions such as some torsion of interface residues or contact numbers could be more descriptive to differentiate different conformations.

Further, what atoms are included in all RMSD calculations involved in this paper? Only alpha carbon or all heavy atoms?

Comments on the Quality of English Language

Improved compared with the previous version. 

Author Response

Reviewer 2 Report (New Reviewer)

Comments and Suggestions for Authors

This is an important study on nematode anticoagulant NAPc2. Simulations suggest a hinge mechanism. Moreover a salt bridge forms that stabilizes the bound simulations. Simulations are nicely complemented with experimental checks.

I have a few suggestions:

-              Fig 2 would look better if the average would have been taken over more simulation runs

-              Line 255: The salt bridge is not observed at all in the NAPc2 255 simulation. How can one be sure the simulation was run long enough? It is stated that the salt bridge is only formed in the later stages of the simulation runs.

Author Response

This manuscript is a resubmission of an earlier submission. The following is a list of the peer review reports and author responses from that submission.

Round 1

Reviewer 1 Report

Comments and Suggestions for Authors

The main point of this paper is that an "internal"  (i.e., intramolecular) salt bridge stabilizes the bound state of the anticoagulant NAPc2 in complex with fXa. The evidence is based on MD simulations and ostensibly confirmed by a gross coagulation assay. I have a few issues. First, "the" NAPc2 NMR structure of the free form of NAPc2 is used. But the NMR result is an ensemble. More members of the ensemble need to be used as a starting point for the simulations. Also, what does AlphaFold say the structure is? What are the statistics of the ion pair partners in the free state (e.g., distance) during the trajectories? Similarly for the complex? The correspondence between Jnh derived from the simulations and measured is actually quite poor and the origin obscured by the averaging effects being non-linear. The idea of stabilization and/or folding on binding is pervasive in the literature and the simulations add little here. What is most interesting is the proposal that the salt bridge is stabilizing. To be true, the authors need to document the distance between the partners and their solvation in the free state and similarly in the complex. The mutant activity data is not persuasive without this information since the authors generated a charge imbalance i.e., if truly "internal" (i.e., buried), then burial of the uncompensated charge will prevent formation of the complex structure. Burial of ion pairs is energetically okay (e.g., calmodulin complexes) but not uncompensated charge. The better mutant choice would have been to eliminate both partners. Finally the assay is too complicated. A direct binding assay (e.g., ITC, fluorescence etc) would be more appropriate and would give real thermodynamic information, which is the point of the paper. Also, working with a binding assay allows a simpler experiment - high salt, which will compete with the formation of the ion pair.

Reviewer 2 Report

Comments and Suggestions for Authors

Dear authors!

Your article makes a good impression and I will recommend it for publication.

I liked your detailed description of the conditions of the computational experiment.

There are a few notes:

In Figure 2, it is advisable to depict a coordinate grid for graphs of RMSD values.

  It is advisable to provide the average RMSD values in the caption to the figure.

The explanation to Figure 4, which shows the correlation matrix, requires more explanation.

about how it was calculated, or you need to provide the appropriate citation.

In your research you rely on the analysis of structural changes in a molecular complex

completely ignoring the change in energies. It seems to me that estimating energy changes and calculating binding energy

of the complex using the MM/GBSA method (included in the AMBER package) would be useful.

All necessary calculations of trajectories (substrate ligand complex) for this have already been performed by you (see Table 1).

Reviewer 3 Report

Comments and Suggestions for Authors

1.- Line 85; Why were the simulations done at 298 K, if the experimental part was done at 315 K?

2.-I recommend making a graph of Etotal vs time of the systems, to justify the time of the molecular dynamics simulation.

3.- In Figure 2, the representations of the secondary structures do not provide much information and confuse the reader.

4.- It would be a good idea to perform a molecular dynamics simulation of the NAPc2 E10Q mutant and compare it with the wt.

Comments on the Quality of English Language

The writing is correct

Reviewer 4 Report

Comments and Suggestions for Authors

Comments on biomolecules-2695695:

For comparison between experiment and computational outcomes, the authors should apply some statistical analyses to quantitatively estimate the consistency. For example, for the Karplus results and the experimental reference in Fig. S1, the authors could use root-mean-squared deviation, max deviation and minimum deviation to evaluate the consistency.

No statistical uncertainty is reported for statistical observables extracted from the simulation trajectories, e.g., RMSF reported in Fig. S1. The authors should make great efforts in this aspect and properly define the fluctuations of all statistical quantities reported in this manuscript.

A lot of writing and formatting issues exist throughout the manuscript. For example, the title of the SI file has typoes, i.e., qSuppoting information. Multiple definitions of fXa could be found in the main article. The authors should carefully proofread their manuscript before submitting it to a journal. The current careless editing is absolutely unacceptable.

The authors have made several observations of the open/close transition of the fXa active site in their simulations. Can they compute the relative stability of the open and closed states?

The Ramachandran plots given in Figure S3 are not really useful. With modern force fields and the initial structure from experiment, visiting favorable regions in the 2D space is almost guaranteed. This analysis is simply useless and provides no novel finding.

A very old force field is applied in this work, ff99SB. There are already many scientific reports up to now identifying the weaknesses of this old parameter set, e.g., Journal of Chemical Information and Modeling 61 (1), 284-297 and Journal of Theoretical and Computational Chemistry 18 (03), 1950015. Modern selections such as ff14SB and ff19SB correct many errors and provide more reasonable protein dynamics. As the length of the simulation trajectories ~100 ns is not long in the current work, the authors are expected to redo some of the simulation and analyses with a more up-to-date force field, e.g., ff14SB.

The sampling interval of configurations is set to a very small value, 200 fs. Such a length has been believed to be too short to decorrelate successive configurations, which could lead to fake convergence and thus systematic biases in the trajectory-averaged estimates.

The font size in all figures should be increased. The current text in plots is unreadable.

Comments on the Quality of English Language

The current careless editing is absolutely unacceptable. The authors should endeavor in this aspect in revision.